# Rossby wave second harmonic generation observed in the middle atmosphere

Maosheng He [1] ✉ & Jeffrey M. Forbes[2] ✉

Second harmonic generation is the lowest-order wave-wave nonlinear interaction occurring in, e.g., optical, radio, and magnetohydrodynamic systems. As a prototype behavior of waves, second harmonic generation is used broadly, e.g., for doubling Laser frequency. Second harmonic generation of Rossby waves has long been believed to be a mechanism of high-frequency Rossby wave generation via cascade from low-frequency waves. Here, we report the observation of a Rossby wave second harmonic generation event in the atmosphere. We diagnose signatures of two transient waves at periods of 16 and 8 days in the terrestrial middle atmosphere, using meteor-radar wind observations over the European and Asian sectors during winter 2018–2019. Their temporal evolution, frequency and wavenumber relations, and phase couplings revealed by bicoherence and biphase analyses demonstrate that the 16-day signature is an atmospheric manifestation of a Rossby wave normal mode, and its second harmonic generation gives rise to the 8-day signature. Our finding confirms the theoretically-anticipated Rossby wave nonlinearity.

Rossby waves (RWs, also known as planetary waves) develop in rotating fluids, owing their existence to the conservation of potential vorticity. The meridional gradient of the Coriolis force resists meridional displacements of flows and drives RWs propagating zonally. Figure 1 sketches RWs' restoring force and phase velocity. In the universe, RWs occur ubiquitously in various astrophysical bodies, e.g., in planets' atmospheres, oceans and liquid cores, e.g., refs. [1–6] and stars' plasma, e.g., refs. [7,8]. The recent observational findings of RWs at the Sun and other astrophysical bodies have promoted a renaissance of studies on RWs, e.g., refs. [9–11].

Transporting massive momentum and energy globally, RWs play a significant role in the transient adjustment of oceanic and atmospheric circulations, e.g., refs. [12,13]. Atmospheric RWs are of importance in determining weather systems on Earth and triggering extreme weather events, e.g., refs. [14–16], whereas oceanic RWs drive climate variability over multiple temporal scales through couplings with the atmosphere, e.g., refs. [17,18]. In addition, the RWs developing on the Sun impact the aerospace plasma environment and play a role in producing space weather events, e.g., refs. [9,10]. Despite these importances, RWs are one of the very rare geophysical phenomena that were predicted theoretically before their observational finding[19]. A similar scenario

also happened relative to the Sun. RWs were unambiguously detected on the Sun[8] decades after their theoretical prediction[20]. The difficulties in observing RWs are owing to the ultra-long temporal and spatial scales. The wave periods, meaning the time a wave takes for two successive crests to pass a specified point, are longer than the astrophysical bodies' rotation periods, while their wavelengths are comparable to the astrophysical bodies' radius. The relevant monitoring or detection entails continuous observations from multiple longitudinal sectors simultaneously in a broad time window. In addition, RWs are often transient, dissipative, and beyond detection. Most detectable RWs are the normal modes associated with atmospheric intrinsic properties. The normal modes are also dissipative and often last only for a few wave periods. Consequently, in the low Earth atmosphere, observational studies on the RW normal modes often require statistical spectral analyses[21,22]. With increasing altitude, amplitudes of atmospheric RW normal modes increase substantially and often maximize in the middle atmosphere, e.g., ref. [23]. Accordingly, the middle atmosphere serves as a natural laboratory for studying RWs and their dissipation mechanisms. A nonlinear behavior of RWs is the second harmonic generation (SHG), e.g., ref. [24], which was predicted numerically in the middle atmosphere[25] and

[1]Key Laboratory of Solar Activity and Space Weather, National Space Science Center, Chinese Academy of Sciences, Beijing, P. R. China. [2]Ann & H.J. Smead Department of Aerospace Engineering Sciences, University of Colorado, Boulder, Boulder, CO, USA. ✉e-mail: hmq512@gmail.com; forbes@colorado.edu

**a** The pressure gradient force balances the Coriolis force at latitude $\varphi_0$

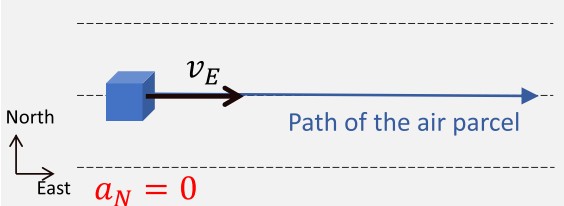

**b** The meridional gradient of Coriolis force resists meridional displacements $\delta\varphi$ when $v_E > 0$

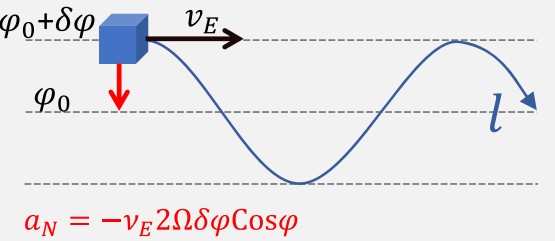

**c** The velocity of the waveform

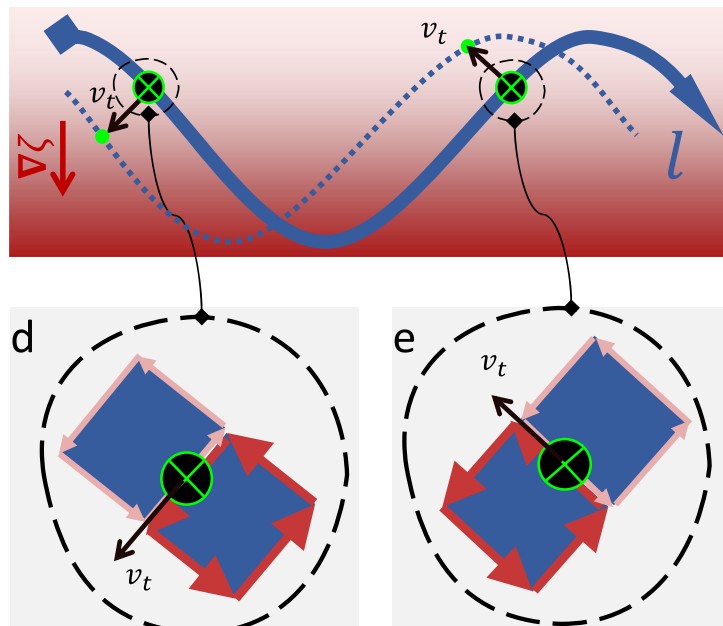

**Fig. 1 | Sketches of Rossby waves' fundamental principles. a, b** The restoring force. **c–e** The waveform's velocity. In (**a**), an air parcel follows along latitude $\varphi_0$ at an eastward velocity $v_E$ with a meridional acceleration $a_N = 0$ when the pressure gradient force balances the Coriolis force. In **b**, when the parcel encounters a small displacement $\delta\varphi$ in latitude, the Coriolis force's gradient imposes a meridional acceleration $a_N = \delta\varphi \mathrm{d}a_C/\mathrm{d}\varphi = -\delta\varphi v_E 2\Omega\mathrm{Cos}\varphi_0$ that always points against $\delta\varphi$ when $v_E > 0$. Here, $\Omega$ denotes the Earth's angular frequency and $a_C = -v_E 2\Omega\mathrm{Sin}\varphi$ is the northward Coriolis acceleration. While the parcel meanders along the blue arrowed line $l$ in (**b**), its waveform travels westward as sketched in **c**. The absolute vorticity composes the planetary vorticity $f = 2\Omega\mathrm{Sin}\varphi$ and the relative vorticity $\zeta$, reflecting the Earth's rotation and the parcel's rotation with respect to the Earth, respectively. The conservation of absolute vorticity $D(\zeta + f)/Dt = 0$ determines a southward gradient of $\zeta$, as denoted by the red shadow in (**c**). The gradient's projection along the flow path $l$ is typically not zero and would cause a tangential velocity $v_t$. As an example, the path $l$ in **c** is zoomed in at two green crosses, displayed in (**d**, **e**). These two crosses are associated with positive and negative gradients of $\zeta$ along $l$, respectively, as denoted by the red and pink arrows in (**d**, **e**). The black arrows $v_t$ denote the vector sums of the red and pink arrows bordering the crosses, both of which project zonally westward. The parcels at these crosses drift toward the green points in (**c**) and, visually, the path $l$ drifts westward toward the dotted line.

theoretically analyzed in the ocean, e.g., refs. [26,27] but has not been reported in actual observations to our best knowledge.

In this work, we use middle atmospheric winds observed over European and Asian sectors at 54–55°N latitude to detect the frequency, zonal wavenumber, and phase couplings of wave signatures appearing in early 2019. Results illustrate that an 8-day wave results from the SHG of a 16-day RW, confirming the theoretically anticipated RW nonlinearity.

## Results

In the middle atmosphere, normal modes can be detected in individual cases. Most of these detections are based on single-station or -satellite approaches, and therefore are subject to inherent spatiotemporal ambiguities, e.g., ref. [28]. To conquer this problem, here we implement a dual-station ground-based approach (see "Methods", subsection "Zonal wavenumber estimation"), employing a cross-wavelet analysis of middle atmospheric horizontal wind observations at 54–55°N latitude from two longitude sectors. Similar to a wavelet spectrum, a cross-wavelet spectrum comprises a complex value as a function of time and frequency and presents extents of perturbations. Different from a wavelet spectrum which depicts the perturbations recorded in a single sensor, a cross-wavelet spectrum indicates perturbations synchronized between two sensors: the complex norm of the cross-wavelet spectrum denotes the products of the amplitudes recorded in each of the sensors, while the complex argument denotes the phase difference between the sensors.

In Fig. 2, the cross-spectrum, from November 2018 to March 2019, is populated by a number of peaks. The six most substantial peaks occur around spectral periods of 16, 4, 2.5, 7, 8, and 6 days, in

descending order of their amplitudes, as indicated by the numbers 1–6 in Fig. 2. The arguments of the spectral peaks reveal that the underlying waves are associated with dominant zonal wavenumbers 1, 2, 2, 2, and 1 (for the wavenumber estimation and the underlying assumptions, see "Methods", subsection "Zonal wavenumber estimation"), respectively, as specified in Table 1. Among these spectral peaks, five are attributable to manifestations of normal modes. As specified in Table 1, the 16-, 4-, 7-, and 6-day peaks are manifestations of the first and second symmetric RW normal modes of zonal wavenumbers 1 and 2[29–31], and the 2.5-day peak is the manifestation of a Rossby-Gravity mode with zonal wavenumber 2[32]. The occurrences of these normal modes are attributable to the seasonality of, e.g., the Rossby-Gravity mode[33], or the associations between stratospheric sudden warming events and the normal modes (e.g., the 16- and 6-day modes)[34,35] in response to the 2019 new year warming event[36]. However, the 8-day spectral peak cannot be attributed to any normal mode, but rather is explicable as the result of a second harmonic generation (SHG) of the 16-day normal mode, as anticipated in the numerical simulation of ref. [25]. Such an RW SHG event is rare. Our investigation explores seven years (2013–2019) of observations and only detects this single significant signature.

The total observed Rossby wave responses at 16 and 8 days can be viewed in terms of the eigenmodes of Laplace's tidal equation that they project onto. Some hints regarding the modes that are present at 55°N can be inferred from the estimation of the vertical wavelength $\lambda_z$ (Supplementary Fig. 2). The range of $\lambda_z$ measured at 55°N for the 16-day wave with zonal wavenumber 1 (41–46 km) is consistent with the presence of the first symmetric and first antisymmetric Rossby modes with $\lambda_z$ of 32 km and 56 km. The 8-day wave with zonal wavenumber 2

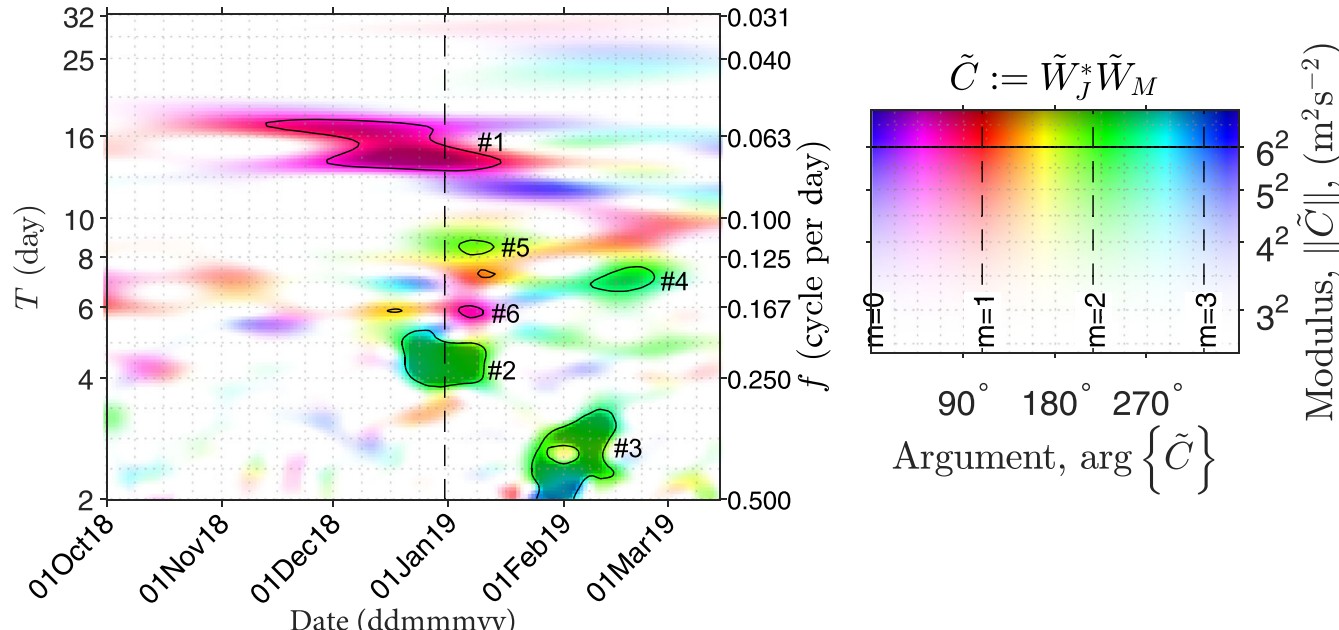

**Fig. 2 | Altitude-averaged (80–96 km) cross-wavelet spectrum of horizontal wind between Mohe (122°E, 54°N) and Juliusruh (13°E, 55°N).** Averaged here is the sum of the spectrum of zonal wind $u$ and that of the meridional wind $v$. The darkness and color hue in each panel denote the modulus and argument of the spectrum, namely, $\|\bar{C}\|$ and $\arg\{\bar{C}\}$, respectively. The phase is a function of zonal wavenumber $m$ as specified in the color code. The six numbers following the symbol # index the most substantial peaks in descending order of their amplitudes, as specified in Table 1. The solid black isolines denote amplitudes at 6 ms$^{-1}$ and the vertical dashed line indicates the central day of the 2019 new year stratospheric sudden warming event. Readers with colour vision deficiencies are referred to Supplementary Fig. 1 for color-filtered versions of the current figure.

range of $\lambda_z$ (64–252 km) is subject to much greater uncertainty, but appears to exclude the first symmetric mode with $\lambda_z$ of 32 km, leaving the first antisymmetric Rossby mode with $\lambda_z$ of 72 km and the second symmetric Rossby near-normal mode with deep vertical scale as possible contributors to the total Rossby wave response at 55°N.

## Discussion

The existence of one wave exhibiting spatial unevenness might affect a second existing wave nonlinearly, resulting in a spatially uneven traveling speed of the second wave and giving rise to a third wave. Mathematically, waves are considered as solutions of linear systems and represented as, e.g.,

$$A_N \psi_N := A_N e^{i2\pi(f_N t - \mathbf{k}_N \cdot \mathbf{r} + \phi_N)} \qquad (1)$$

Here, $A$ and $\psi$ denote the absolute amplitude and phase of the wave; $f$, $\mathbf{k}$, and $\phi$ denote the wave frequency, wavenumber and initial phase; $t$ and $\mathbf{r}$ denote time and position; and $N$ is the index of solutions.

**Table 1 | Spectral peaks in Fig. 2 and their interpretations**

| index | date | $T^a$ | $m^b$ | interpretation | reference |
|---|---|---|---|---|---|
| #1 | Nov.–Jan. | 14–18 | 1 | AM$^c$ of the 2nd symmetric normal mode of $m = 1$ | [67] |
| #2 | Dec.–Jan. | 4–5 | 2 | AM of the 1st symmetric normal mode of $m = 2$ | [30] |
| #3 | Jan.–Feb. | 2–3 | 2 | AM of a Rossby-Gravity mode | [32] |
| #4 | Feb. | 7 | 2 | AM of the 2nd symmetric normal mode of $m = 2$ | [31] |
| #5 | Jan. | 8–9 | 2 | AM of the SHG of the wave in #1 | [25] |
| #6 | Jan. | 5–6 | 1 | AM of the 1st symmetric normal mode of $m = 1$ | [30] |

$^a T$ denotes the spectral period in the unit of day.
$^b m$ denotes the zonal wavenumber in the unit of cycle per 360° longitude.
$^c$ AM abbreviates 'atmospheric manifestation'.

If a linear system has two solutions $\psi_1$ and $\psi_2$, their linear combination $A_1\psi_1 + A_2\psi_2$ is also a solution, according to the superposition principle. In a weakly nonlinear system, the superposition solution requires a correction with quadratic terms: $A_1\psi_1 + A_2\psi_2 + \sum_{i,j\in\{1,2\}} A_{i,j}\psi_i\psi_j$. The correction enables the propagation of three potential waves $\psi_{i,j} := \psi_i\psi_j$ (specifically, $\psi_{1,2} := \psi_1\psi_2$, $\psi_{1,1} := \psi_1\psi_1$, and $\psi_{2,2} := \psi_2\psi_2$). Such a generation of $\psi_{i,j}$ are called, e.g., wave–wave nonlinear interaction or weakly wave interaction. The definition of $\psi_{i,j}$ implies a restrictive phase-matching among the forcing parent waves and the generated child wave: $\psi_{i,j}^*\psi_i\psi_j \equiv 1$, or,

$$\arg\left\{\psi_{i,j}^*\psi_i\psi_j\right\} \equiv 0 \qquad (2)$$

or,

$$\begin{cases} f_{i,j} = f_i + f_j \\ \mathbf{k}_{i,j} = \mathbf{k}_i + \mathbf{k}_j \\ \arg\left\{e^{i2\pi(\phi_{i,j} - \phi_i - \phi_j)}\right\} = 0, \text{ or, } \phi_{i,j} - \phi_i - \phi_j \in \mathbb{Z} \end{cases} \qquad (3)$$

The phase-matching relations Eqs. (2) and (3) are often discussed under the constraining of non-negative frequencies, $f_i, f_j \in \mathbb{R}_{\geq 0}$, e.g., in refs. [37,38], which in principle hold for any real frequencies, $f_i$, $f_j \in \mathbb{R}$. Under real frequency constraint, Eqs. (3) imply $\|f_{i,j}\| = \|f_i\| + \|f_j\|$ or $\|f_{i,j}\| = \|\|f_i\| - \|f_j\|\|$ (see "Methods", subsection "Notation of wave–wave interactions"). For notational convenience, here we constrain frequencies to be non-negative $f \in \mathbb{R}_{\geq 0}$ to consider only the case $\|f_{i,j}\| = f_{i,j} = f_i + f_j$.

In Eq. (2), $\psi_{i,j}$ corresponds to the interaction between the $i$- and $j$-th forcing parent waves. Specially, when $i = j$, $\psi_{i,i}$ corresponds to the quadratic nonlinearity of self-interaction, known as the second harmonic generation (SHG). In SHG, Eqs. (3) read $f_{i,i} = 2f_i$, $\mathbf{k}_{i,i} = 2\mathbf{k}_i$, and $e^{i2\pi(\phi_{i,i} - 2\phi_i)} = 1$. Satisfying the two relations are the signatures in Fig. 2, where both spectral frequency and zonal wavenumber of the 8-day

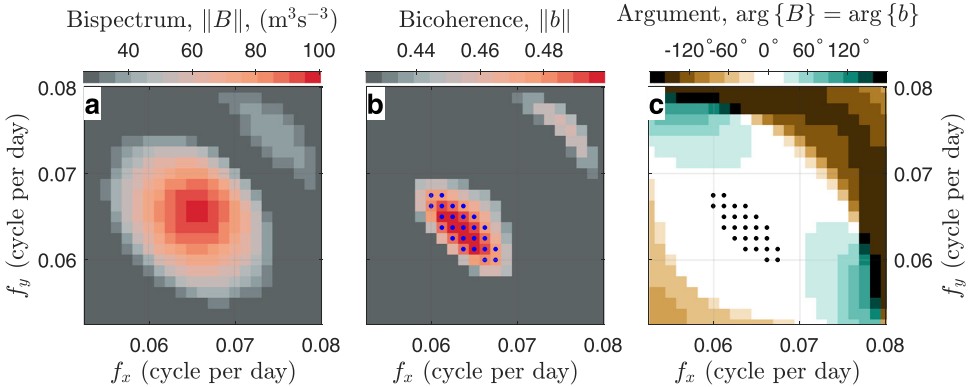

**Fig. 3 | Bispectral spectra averaged within 80–96 km, between Juliusruh and Mohe, and between zonal and meridional winds observed between 1 November 2018 and 31 January 2019. a** Bispectrum. **b** Bicoherence. **c** Bispectral argument (namely, biphase). In (**b**) and (**c**), the dots indicate the significance level above $\alpha = 0.01$.

peak are twice those of the 16-day peak. Therefore, we conclude that the 8-day peak is a signature of SHG. Further factors support this conclusion. The 8-day peak starts to enhance when the 16-day peak starts to weaken. This anti-correlated temporal evolution is attributable to the energy transport from the 16-day wave to the secondary wave (see "Methods", subsection "Energy conversation in wave–wave interactions").

While the anti-correlation reflects the energy budget of the interactions, a complex correlation defined between energy-supplying and -receiving parties reflects the coherence among involved waves (see "Methods", subsection "Bispectrum and bicoherence"). In the case of quadratic nonlinearity, which typically involves three waves, the complex correlation is defined between the complex amplitude of the highest frequency wave and the product of the complex amplitudes of the other two waves. If $\mathcal{X}(f)$ denotes the complex amplitude of oscillation at frequency $f$ estimated spectrally from an time series, the covariance and correlation coefficient between the variables $\mathcal{X}(f_1 + f_2)$ and $\mathcal{X}(f_1)\mathcal{X}(f_2)$ are the bispectrum and bicoherence, respectively. Triple oscillations at three frequencies $f_1$, $f_2$ and $f_1 + f_2$ ($\langle \parallel \mathcal{X}(f_1) \parallel \rangle > 0$, $\langle \parallel \mathcal{X}(f_2) \parallel \rangle > 0$, and $\langle \parallel \mathcal{X}(f_1 + f_2) \parallel \rangle > 0$) excited spontaneously by independent waves or processes with random phases are characterized by a near-zero bicoherence. In contrast, the significant bispectral and bicoherence peak (Fig. 3a, b) suggests coherence among the oscillations.

The bispectral peak is associated with a near-zero argument (Fig. 3c). The near-zero argument implies that the 16-day wave peak overlaps with the 8-day wave peak in space. This overlap satisfies the phase-matching relation of the initial phase in the last equation in Eq. (3). Compared to the phase-matching relations of wave frequency and wavenumber, the initial phase condition has rarely been discussed in the existing literature. One potential reason is that the initial phase condition pertains to wave components whose energy is completely exchanged through the interaction. When any wave participates in the interaction partially and the initial phase of the interacting part is different from that of the remaining part, the frequency and wavenumber conditions are still observable, but the initial phase condition will not be.

The nonlinear interactions and SHG broaden spectral variability by cascading and diluting energy across discrete spatial-temporal scales, either upscale or downscale. As a prototype wave behavior, SHG occurs in various systems and is broadly used, for instance, in nonlinear optics and radio science, e.g., refs. [24,39,40]. Numerical simulations of SHGs of atmospheric Rossby wave normal modes date back to refs. [25,41]. Here, we present observation of an unambiguous event of the Rossby wave SHG, under the constraints of both wave frequency and zonal wavenumber as well as the triplet coherence.

Rossby waves and their nonlinear interactions play important roles in shaping the weather of atmospheres, oceans, and plasma at Earth, Sun and other astrophysical bodies[9,11,42], which are also used to interpret various intriguing problems and astrophysical periodicities, such as the solar 22-year Hale cycle and 11-year Schwabe cycle[43,44], quasi periodicities in solar surface magnetic structures in differential rotation and toroidal field amplitudes[45], and secular variations of the geomagnetic field[2,3,6,46]. Therefore, our finding should shed light on the applicability of RW SHG to these intriguing problems.

## Methods
The current work uses middle atmospheric observations from two longitudes at the same latitude to diagnose the zonal wavenumbers of Rossby waves through a dual-station method. Bispectral analysis is used for identifying wave–wave nonlinear interactions.

### Meteor wind observations
We use two meteor radars from two longitude sectors, at Mohe (122°E, 54°N) and Juliusruh (13°E, 55°N) between June 2018 and June 2019. For the radar setups, e.g., operating radio frequencies and antenna configurations, readers are referred to[47] and[48]. We estimate the hourly zonal and meridional winds ($u$ and $v$) at altitudes $h = 80.5, 81.5,...,99.5$ km at each station. The data availability is specified in Section Data availability.

### Zonal wavenumber estimation
Oscillations induced by one traveling wave are coherent everywhere on the wave's path. The phase difference between the two sites is time-invariant and proportional to the spatial separation times the wavenumber in the detection defined by the two sites, which provides an opportunity to diagnose the directional wavenumber experimentally. According to this principle,[49] developed a dual-station approach, called the phase differing technique (PDT), for diagnosing zonal wavenumber using observations from two zonally separated stations.

A zonally traveling wave with zonal wavenumber $m$ and frequency $f$ can be denoted as $\bar{\Psi}(\lambda, t \,|\, f, m) = \tilde{A}e^{i(2\pi f t + m\lambda)} := \tilde{a}(\lambda \,|\, m)e^{i2\pi f t}$, where $t$ and $\lambda$ represent time and longitude; $\tilde{A}$ represents the wave's complex amplitude and $\tilde{a} := \tilde{A}e^{im\lambda}$. We define a cross-product: $\tilde{c} := \tilde{a}(\lambda_1 \,|\, m)\tilde{a}^*(\lambda_2 \,|\, m) = \parallel \tilde{A} \parallel^2 e^{im(\lambda_1 - \lambda_2)}$ where $\lambda_1$ and $\lambda_2$ denote two longitudes. Accordingly,

$$m = \frac{\arg\{\tilde{c}\} + 2\pi Z}{\lambda_1 - \lambda_2} \tag{4}$$

Here, $Z \in \mathbb{Z}$ is an integer, $\arg\{\tilde{c}\} + 2\pi Z$ denotes the phase difference of the wave between $\lambda_1$ and $\lambda_2$, and $\frac{2\pi Z}{\lambda_1 - \lambda_2}$ corresponds to the Nyquist

wavenumber $m_N := \frac{\pi}{\lambda_1 - \lambda_2}$. Assuming that at a given latitude and at a given wave frequency, there exists one dominant wave traveling zonally, we can estimate $\tilde{a}(\lambda_1|m)$, $\tilde{a}(\lambda_2|m)$, and $\tilde{c}$ experimentally through spectral analysis of the time series of observations collected at $\lambda_1$ and $\lambda_2$.

Following[50] and using a similar radar configuration, at Mohe (M) and Juliusruh (J), we first calculate the wavelet spectra of the zonal and meridional wind components $u$ and $v$ over each station at each altitude $h$, resulting in amplitude spectra $\tilde{W}_{J,u,h}$, $\tilde{W}_{J,v,h}$, $\tilde{W}_{M,u,h}$, and $\tilde{W}_{M,v,h}$. We sum the cross-products of the two components at each altitude, $\tilde{C}_{u+v} := \tilde{W}^*_{J,u}\tilde{W}_{M,u} + \tilde{W}^*_{J,v}\tilde{W}_{M,v}$, and average the sum in the altitude range between 80 and 96 km, resulting in $\langle\tilde{C}_{u+v}\rangle_{80<h<96km}$ as displayed in Fig. 2. $\tilde{W}$ and $\langle\tilde{C}\rangle$ are the experimental estimations of $\tilde{a}$ and $\tilde{c}$ in Eq. (4).

The estimation of $m$ according to Eq. (4) is subject to the aliasing associated with the Nyquist wavenumber $m_N = \frac{\pi}{\lambda_1 - \lambda_2}$, namely, the $\|m\|$ of the underlying wave should be smaller than $\|m_N\|$ or in Eq. (4) (or, $Z = 0$, or the underlying wavelength (namely, the distance between two consecutive wave crests or between two consecutive wave troughs) is longer than twice the station separation). This long-wave assumption $Z = 0$ could be relaxed slightly to $Z \in \{-1, 0, 1\}$, under the assumptions that the underlying wavenumber is a near-zero integer $m \in \{-1, 0, 1, 2, 3\}$. Following[35], we determine $m$ through the following optimization,

$$\hat{m} = \underset{Z\in\{-1,0,1\}, m\in\{-1,0,1,2,3\}}{\operatorname{argmin}} \| \arg\{\langle\tilde{C}_{u+v}\rangle\} + 2\pi Z - m(\lambda_M - \lambda_J) \| \quad (5)$$

This approach has been evaluated through comparisons with estimations from different radar pair configurations at the same latitude and estimations through least-square fit using observations from three or four longitudes[33,51].

## Notation of wave–wave interactions

In "Discussion", we constrained frequencies to be non-negative in discussing the quadratic interaction. For completeness, here we use real-value frequencies to introduce the interaction.

According to Eq. (1), $A_N\psi_N$ and its conjugate $A_N\psi^*_N$ denote waves with opposite polarizations in the complex plane. These two waves share the same real part $\Re\{\psi_N\} \equiv \Re\{\psi^*_N\}$ and therefore both exhibit experimentally as the same signal $A_N\Re\{\psi_N\}$. Generally, the experimental signal $\Re\{\psi_N\}$ represents a linear combination of two waves $\Psi_N := (p_N \cdot \psi_N + (1 - p_N) \cdot \psi^*_N)$, since $\Re\{\Psi_N\} \equiv \Re\{\psi_N\}$. Here the $p_N \in \mathbb{R}$ is an arbitrary real number. The interaction between two experimental signals $\Re\{\psi_i\}$ and $\Re\{\psi_j\}$ involves $\Psi_i$ and $\Psi_j$ and the denotation $\psi_{i,j} := \psi_i\psi_j$ used in "Discussion" should be generalized as $\Psi_{i,j} := \Psi_i\Psi_j = p_ip_j\psi_i\psi_j + (1 - p_i)p_j\psi^*_i\psi_j + p_i(1 - p_j)\psi_i\psi^*_j + (1 - p_i)(1 - p_j)\psi^*_i\psi^*_j$. Using denotations of matrix multiplication, $\Psi_{i,j} = \underline{P}^t\underline{\psi}$, where $\underline{P}^t := [p_ip_j, (1 - p_i)p_j, p_i(1 - p_j), (1 - p_i)(1 - p_j)]$, $\underline{\psi}^t := [\psi_i\psi_j, \psi^*_i\psi_j, \psi_i\psi^*_j, \psi^*_i\psi^*_j]^t$. Here, the underlined letters denote column vectors, and the superscript $t$ denotes transpose of a vector. The linear combination $\Psi_{i,j} = \underline{P}^t\underline{\psi}$ represents that the interaction between $\Psi_i$ and $\Psi_j$ might generate at maximum four quadratic terms. For convenience, we denote these terms as $\psi_{+i,+j} := \psi_i\psi_j$, $\psi_{-i,+j} := \psi^*_i\psi_j$, $\psi_{+i,-j} := \psi_i\psi^*_j$, and $\psi_{-i,-j} := \psi^*_i\psi^*_j$. Their phase matching could be summarized as

$$\begin{cases} f_{\pm i, \pm j} = \pm f_i \pm f_j \\ \mathbf{k}_{\pm i, \pm j} = \pm \mathbf{k}_i \pm \mathbf{k}_j \\ \phi_{\pm i, \pm j} \mp \phi_i \mp \phi_j \in \mathbb{Z} \end{cases}, \text{ and } \begin{cases} f_{\mp i, \pm j} = \mp f_i \pm f_j \\ \mathbf{k}_{\mp i, \pm j} = \mp \mathbf{k}_i \pm \mathbf{k}_j \\ \phi_{\mp i, \pm j} \pm \phi_i \mp \phi_j \in \mathbb{Z} \end{cases} \quad (6)$$

These four terms could yield only two independent experimental wave signals: $\Re\{\psi_{\pm i, \pm j}\}$ and $\Re\{\psi_{\pm i, \mp j}\}$ due to $\Re\{\psi_i\psi_j\} \equiv \Re\{\psi^*_i\psi^*_j\}$ and $\Re\{\psi_i\psi_j\} \equiv \Re\{\psi_i\psi^*_j\}$. These signals occur at two absolute frequencies

that can be denoted as a set $\{\|f_{\pm i, \pm j}\|, \|f_{\mp i, \pm j}\|\} = \{\|f_i\| + \|f_j\|, \|\|f_i\| - \|f_j\|\|\}$.

The indices $i, j \in \{1, 2\}$ are defined in "Discussion", implicating three possible combinations, namely, $\{i, j\} = \{1, 1\}$, $\{2, 2\}$, or $\{1, 2\}$. These possibilities are associated with six independent experimental wave signals, namely, $\Re\{\psi_{\pm 1, \pm 1}\}$, $\Re\{\psi_{\mp 1, \pm 1}\}$, $\Re\{\psi_{\pm 2, \pm 2}\}$, $\Re\{\psi_{\mp 2, \pm 2}\}$, $\Re\{\psi_{\pm 1, \pm 2}\}$, and $\Re\{\psi_{\mp 1, \pm 2}\}$. Among them, $\Re\{\psi_{\mp 1, \pm 1}\}$ and $\Re\{\psi_{\mp 2, \pm 2}\}$ are trivial solutions (zero-frequency and zero-wavenumber), while the others occur at four absolute frequencies, $\|2f_1\|$, $\|2f_2\|$, $\|f_1\| + \|f_2\|$, and $\|\|f_1\| - \|f_2\|\|$, known as the four possible secondary waves of quadratic interactions between wave signals $\Re\{\psi_1\}$ and $\Re\{\psi_2\}$[e.g.,25]. These secondary waves might occur independently from each other, e.g., ref. [52], and the interaction might occur between different types of waves, e.g., ref. [53]. The secondary waves at frequencies $\|f_1\| + \|f_2\|$ and $\|\|f_1\| - \|f_2\|\|$ are known as upper and lower sidebands (USB and LSB). The generation of waves at $\|2f_1\|$ and $\|2f_2\|$ corresponds to the SHG. SHG is a special USB generation in which the roles of both parent waves are played by the same wave.

Since all negative-frequency wave solutions can be denoted notationally as conjugations of positive-frequency solutions, frequencies are often constrained to be non-negative $f \in \mathbb{R}_{\geq 0}$, as used in "Discussion".

## Bispectrum and bicoherence

Bispectrum measures a higher-order moment of a time series, developed first for investigating oceanic waves[54]. Existing literature proposed and named at least three unitless normalizations of bispectrum as bicoherence, e.g., refs. [55–57]. A comparison[58] favors a normalization, e.g., ref. [55] bounded between 0 and 1. The current work uses this normalization and proposes a new interpretation for it. This normalization could be interpreted as the correlation coefficient of two particular complex variables, $U := \mathcal{F}(f_1)\mathcal{F}(f_2)$ and $V := \mathcal{F}(f_1 + f_2)$. The covariance of these two variables is an interpretation of the bispectrum. Here, $\mathcal{F}(f)$ denotes the complex Fourier amplitude of a parameter $x$ at spectral frequency $f > 0$ and position $\mathbf{r}$ within a time window centering at $t$. In the occurrence of a nonlinear interaction involving three waves, $U$ and $V$ represent the energy-supplying and -receiving parties, and their correlation,

$$\text{Cov}(U, V) = \langle(U - \langle U\rangle)(V - \langle V\rangle)^*\rangle \quad (7)$$

is facilitated by the energy conversation (see "Methods", subsection "Energy conversation in wave–wave interactions"). Suppose the time and spatial variation of $x$ can be denoted as a linear combination of plane waves, namely, $x(t, \mathbf{r}) = \sum_n A_n\psi_n(\lambda, t|f, \mathbf{k})$ where $\psi_n = e^{i2\pi(f_nt - \mathbf{k}_n\cdot\mathbf{r} + \phi_n)}$ denotes the $n$-th wave. For any $f_1$, the expectation $\langle\mathcal{F}(f_1)\rangle_t = 0$, because

$$\langle\mathcal{F}(f_1)\rangle_t = \frac{\int_{t_0}^{t_0 + \Delta T} A_1\psi_1 dt}{\Delta T} = \frac{A_1e^{i2\pi(-\mathbf{k}_1\cdot\mathbf{r} + \phi_1)}}{i2\pi f_1\Delta T}e^{i2\pi f_1 t}|_{t_0}^{t_0 + \Delta T}$$
$$\Rightarrow \|\langle\mathcal{F}(f_1)\rangle_t\| \leq \frac{\|A_1\|}{\pi f_1\Delta T} \quad (8)$$
$$\Rightarrow \lim_{\Delta T \to +\infty} \|\langle\mathcal{F}(f_1)\rangle_t\| = 0$$

Similarly, $\langle\mathcal{F}(f_1)\rangle_\mathbf{r} = 0$, $\langle\mathcal{F}(f_1)\rangle_{t,\mathbf{r}} = 0$ and $\langle\mathcal{F}(f_1)\mathcal{F}(f_2)\rangle_{t,\mathbf{r}} = 0$ at any spectral frequencies $f_1$ and $f_2$. Therefore, $\langle U\rangle = 0$ and $\langle V\rangle = 0$. Substitute these expectations into Eq. (7), yielding,

$$\text{Cov}(U, V) = \langle UV^*\rangle = \langle\mathcal{F}(f_1)\mathcal{F}(f_2)\mathcal{F}(f_1 + f_2)^*\rangle := B(f_1f_2) \quad (9)$$

 5

Similarly, the correlation coefficient between $U$ and $V$ reads,

$$\begin{aligned}
\rho_{U,V} &= \frac{\mathrm{Cov}(U,V)}{\sqrt{\langle \parallel U - \langle U \rangle \parallel^2 \rangle \langle \parallel V - \langle V \rangle \parallel^2 \rangle}} \\
&= \frac{B(f_1 f_2)}{\sqrt{\langle \parallel U \parallel^2 \rangle \langle \parallel V \parallel^2 \rangle}} \\
&= \frac{B(f_1 f_2)}{\sqrt{\langle \parallel \mathcal{F}(f_1)\mathcal{F}(f_2) \parallel^2 \rangle \langle \parallel \mathcal{F}(f_1+f_2) \parallel^2 \rangle}} \\
&:= b(f_1 f_2)
\end{aligned} \tag{10}$$

$B(f_1, f_2)$ and $b(f_1, f_2)$ were defined as the bispectrum and bicoherence of $x(t, \mathbf{r})$, respectively[55]. A near-zero $\parallel b(f_1,f_2) \parallel$ reveals a random phase mixing of the three oscillations, $\mathcal{F}(f_1)$, $\mathcal{F}(f_2)$, and $\mathcal{F}(f_1+f_2)$, suggesting these oscillations might be spontaneously and independently excited. On the contrary, significant $B(f_1,f_2)$ and $\parallel b(f_1,f_2) \parallel$ values suggest that the three oscillations are coherent with each other. Quantifying the goodness of the coherence, $\parallel b(f_1,f_2) \parallel$ is bounded by $0 \le \parallel b \parallel \le 1$. An explanation for the triplet coherence is a nonlinear interaction among these three oscillations, which is enabled by the phase-matching relations in Eq. (2). Consequently, significant $B(f_1,f_2)$ and $\parallel b(f_1,f_2) \parallel$ are broadly used as experimental indicators of wave–wave nonlinear interaction, e.g., ref. [59]. The argument of Bispectra and bicoherence $\arg\{B\} = \arg\{b\} = 0$ is known as biphase, reflecting the phase relations among the involved waves and implicating the skewness and waveform of the supposed interacting waves, e.g., ref. [60].

Through a Lomb-Scargle analysis, we first estimate the complex amplitude of each wind component $u$ and $v$ over each station between 1 November 2018 and 1 February 2019, at each altitude level. Then, the Bispectrum $\parallel B \parallel$ and bicoherence $\parallel b \parallel$ are calculated according to Eqs. (9) and (10) where averages are across all altitude levels, between $u$ and $v$ and between Mohe and Juliusruh. $\parallel B \parallel$, $\parallel b \parallel$ and their argument, $\arg\{B\}$ are displayed in Fig. 3a–c, respectively.

### Energy conversation in wave–wave interactions

The energy budget of wave–wave interactions is regulated by the Manley–Rowe relations[61] that in each interaction, the energy exported from or accepted by each wave is proportional to the wave's absolute frequency. Under this regulation, the frequency matching $f_{i,j} = f_i + f_j$ in Eq. (3) is equivalent to energy conversation. This implies that the energy is exported either from the highest frequency wave at $f_{i,j}$ to the other two waves at $f_i$ and $f_j$ or on the other way around from waves at $f_i$ and $f_j$ to the wave at $f_{i,j}$. In both cases, the highest frequency wave is one party in the energy budgets, either receiving or exporting energy, and the other two waves are the other party. These two parties are represented by the variables $V$ and $U$, respectively, as defined in "Methods", subsection "Bispectrum and bicoherence". The energy exchange between these parties implies their anti-correlation and facilitates the complex correlation between $V$ and $U$ (namely, the bicoherence as explained in "Methods", subsection "Bispectrum and bicoherence").

In the USB generation, both parent waves export energy to the USB, whereas in the LSB generation, energy is transported from the highest frequency parent wave to both the other parent wave and the LSB. The amplified parent wave was called anti-wave, e.g., ref. [62], which stimulates the occurrence of the LSB interaction and amplifies the Rossby wave, e.g., [52]. This anti-wave is characterized by a negative frequency in the notations defined in "Methods", subsection "Notation of wave-wave interactions". There is a special case in which two initial waves interact, producing a USB, and then the USB interacts with the initial waves amplifying the initial waves. Energy exchanges between the initial waves and the USB back and forth periodically. Such a period is called the modulation period, e.g., ref. [63], which is beyond the scope of the current work.

## Data availability

The hourly wind data at Mohe is available in the word data center (WDC) for Geophysics, Beijing, with the identifier https://doi.org/10.12197/2020GA016[64]. The hourly wind data at Juliusruh is available in the service RADAR, with the identifier https://doi.org/10.22000/343[65]. The datasets generated during and/or analyzed during the current study are available from the corresponding author upon request.

## Code availability

The current work uses the MATLAB Signal Processing Toolbox and Wavelet software provided by C. Torrence and G. Compo[66]. The Wavelet software is available at the http://atoc.colorado.edu/research/wavelets/.

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

## Acknowledgements

This work was supported in part by National Science Foundation Award AGS-1630177 (JMF) to the University of Colorado at Boulder. The wind data at Mohe and Juliusruh that support the findings of this study are provided by National Earth System Science Data Sharing Infrastructure at BNOSE (Beijing National Observatory of Space Environment), IGGCAS (Institute of Geology and Geophysics, Chinese Academy of Sciences) and Leibniz Institute of Atmospheric Physics at the University of Rostock, respectively. The authors thank Professor Dehai Luo (Institute of atmospheric physics, Chinese academy of sciences, Beijing, China) and Professor Antonio Speranza (Institute of Atmospheric Sciences and Climate, Bologna, Italy) for their discussions.

## Author contributions

M.H. and J.M.F. conduct the investigation and formal analysis and write and revise the paper.

## Competing interests

The authors declare no competing interests.
