## [Peer Review File · Nature Communications]

Editorial Note: Parts of this peer review file have been redacted as indicated to avoid any copy right infringement.

REVIEWER COMMENTS

Reviewer #1 (Remarks to the Author):

This paper considers the observation of second harmonic generation of Rossby waves in the terrestrial atmosphere.

The authors used two meteor radars and found the wave frequencies and wavenumbers through a dual-station method.

The analysis showed that the low-frequency 16-day wave was followed by higher frequency 8-day wave with twice wavenumber. Therefore, the authors consider the high frequency mode as due to quadratic nonlinearity of self-interaction. This is an interesting work, which could be of importance for the terrestrial atmosphere as well as for the general theory of Rossby waves. Therefore, I will recommend the publication of the paper after several clarifications (see below), which may enhance the value of the paper.

1. The authors use the resonant conditions of multi-wave nonlinear interaction to explain their observations. Namely, they suggest the process of SHG with conditions of frequency ($f_{i,i}=2f_i$), wavenumber ($k_{i,i}=2k_i$), and phase relations. General wave-wave interaction requires that all waves (initial and resulted) must satisfy the corresponding dispersion relations (for a two-wave interaction see e.g. Zaqarashvili and Roberts, 2006). However, it is hard to be matched in the case of Rossby waves. The authors should show that the both waves satisfy the dispersion properties of Rossby waves. It is clear that 16-day initial wave satisfies the dispersion relation as second symmetric normal mode with $m=1$, but the similar discussion is absent for the SHG.

2. The vertical structure of the wave modes is of vital importance to estimate the equivalent depth and consequently dispersion relations of the waves. The authors mentioned that the power spectrum is altitude-averaged. Is it possible to make, at least rough, estimation of the wave vertical structure and hence the equivalent depth? This may help the authors to answer the question 1.

3. The wave amplitudes are important parameters in nonlinear interactions. The estimation of ratio between wave amplitude and phase speed of 16-day and 8-day waves may allow the authors to estimate the energy transfer time scale from 16 to 8-day oscillations. Then one can compare it to the observed time difference between the modes. This will enhance the confidence of reader in the interpretation suggested by the authors.

4. Wave-wave interaction in the solar dynamo layer (references 5-6) is not yet confirmed to be a true process (though it is interesting). Therefore, I would advise to remove the discussion from the abstract but keep it in introduction. But, of course, it is up to the authors.

Reviewer #2 (Remarks to the Author):

This paper reports a first observation of the excitation of a second harmonic of a Rossby wave in the middle atmosphere, in particular a wave with period ~ 16 days nonlinearly exciting one by resonance with half that period.

This observation is certainly interesting and worth reporting. The analysis methods are given in some detail, and the literature references are rather comprehensive. The authors well recognize the many places in the universe Rossby waves should be present.

That said, there are several ways the paper should be revised and improved before it is ready for publication.

1. The analysis terminology is very 'signal processing' oriented, as in EE, and, as such, will be difficult for geo and astrophysical readers to follow. More common language with brief intuitive descriptions understandable by a wider audience need to be included. Examples: 'cross-wavelet', 'bicoherence'

2. More importantly, a first observation is claimed here, but there is little evidence in the text that the authors have checked the relevant literature for previous examples. It is not clear that these authors know the lower atmosphere literature well enough to be confident in this claim. It may be on more solid ground by comparison with other middle atmosphere analyses. Authors need to document how hard they looked, across what range of literature. It seems likely this phenomenon has been seen in the troposphere; Rossby waves there have been actively studied for at least 80 years.

3. Related to item 2, for readers to judge how important this observation is, they need to be given a better idea of how frequently such SHGs are likely to be present and detectable. Have the authors scanned a much longer sequence of synoptic maps or other data to estimate the frequency of

occurrence? They need to present evidence of that. Detailed analysis of other cases is not necessary for this first study, but please give the reader some idea of how many times this behavior is likely to be found in much longer records. The importance of this behavior for simulation and especially prediction is tied to its frequency of occurrence.

4. A common problem with language used in analyzing naturally occurring waves has to do with terminology like 'periods', 'frequencies' and 'wavelength'. 'Period' can mean the time it takes for an Rossby wave to propagate by its own wavelength, or how long it takes to propagate all the way around the spherical object, or it could relate to a periodic variation in the amplitude of the wave. The latter is particularly relevant when nonlinear couplings are present. The authors have certain definitions of these words in mind, and they may be consistent in their descriptions, but all these features are present in the data, so they need to distinguish carefully among them. 'Resonance' usually refers to an unusually large induced amplitude as well as period. Is the resonance reported here itself periodic in amplitude?

5. On a more detailed point, the identification of various types of modes discussed in lines 74-88 is too glib to be totally convincing; this discussion is key to the subsequent analyses. The words used, namely 'suggest', 'can be', 'might be' suggest the authors themselves have their doubts. More clarity on the uncertainty is needed here.

6. There are places where the language does not make sense, or is scientifically inaccurate:

line 41: 'weathering' probably should be 'weather in'

line 51-53: the statement about detectability of Rossby waves with tides and gravity waves present applies perhaps in the upper or middle atmosphere, but not in the troposphere, where tides are much harder to detect in velocities compared to Rossby waves, and gravity waves are much more 'patchy' in their occurrence.

line 68: what is 'round-based approach'? Probably should be 'ground-based approach'?

7. In the solar context, the recently discovered 'tachocline nonlinear oscillations' (TNOs) should be mentioned, which also involve nonlinear mode mode interactions between Rossby waves of specific longitude wave number with themselves to produce changes in differential rotation and toroidal field ($m=0$ modes) amplitudes (Dikpati, Cally, McIntosh & Heifetz 2017, NatSR).

This reviewer recommends revisions of the paper with attention to the points raised above. Happy to review the revised version.

Reviewer #3 (Remarks to the Author):

Review of the paper

“Discovery of Rossby wave second harmonic generation”

by He and Forbes.

The manuscript, of course, is relevant, and may be of interest to a wide range of specialists in the field of not only atmospheric physics, but also, for example, astrophysics and even hydrodynamics.

Nowadays, not as many studies (as we would like to), are dedicated to the study of the generation of secondary (tertiary) waves in the atmosphere due to nonlinear interactions. In particular, similar studies are being carried out by our working group, with consideration of the balance equation of perturbed potential enstrophy, taking into account the rejection of the quasi-geostrophic approximation.

The work certainly deserves to be published.

However, I would like to note a few points to which the authors could pay attention.

Firstly, I am not very familiar with the requirements of this journal, but the structure of the work seems strange to me. First, the results and conclusions are presented, and then the methods and approaches. Usually, it's the other way around. This makes it somewhat difficult to understand.

Next, I will refer to specific lines:

27: The topic of solar and geomagnetic activity is not disclosed in the article. Therefore, I see no reason to announce it in the abstract.

35: I would like to take a closer look at the theoretical foundations of Rossby waves (and their importance for atmospheric dynamics), so that it is clear to a wide range of readers. Maybe even add a picture or scheme.

51: weak amplitudes – this is only about troposphere.

106: As it was shown by (e.g., Longuet-Higgins, 1968, Pogoreltsev, 2001 – references are in the manuscript), if a signal consisting of two cosine waves with zonal wavenumbers and frequencies (m_1, a_1) and (m_2, a_2) pass through some quadratic system, the output from this system will contain the secondary waves $(2m_1, 2f_1)$, $(2m_2, 2f_2)$, $(m_1 + m_2, f_1 + f_2)$ and $(m_1 - m_2, f_1 - f_2)$. Eq (2) shows only summing. Or I didn't understand something.

In that context, you can also discuss the possible generation of an 8-day wave by other waves, for example, 5-day and 13-day ($m=1$), which is visible in your Fig.1

160: capture to Fig.1 remove "(a)" or add "a,b" to the Figure.

Also, it would be much more convenient if you put Figure 1 and Table 1 as close to each other as possible.

Reviewer #1 (Remarks to the Author):

This paper considers the observation of second harmonic generation of Rossby waves in the terrestrial atmosphere.

The authors used two meteor radars and found the wave frequencies and wavenumbers through a dual-station method.

The analysis showed that the low-frequency 16-day wave was followed by higher frequency 8-day wave with twice wavenumber. Therefore, the authors consider the high frequency mode as due to quadratic nonlinearity of self-interaction. This is an interesting work, which could be of importance for the terrestrial atmosphere as well as for the general theory of Rossby waves. Therefore, I will recommend the publication of the paper after several clarifications (see below), which may enhance the value of the paper.

We appreciate your insightful feedback, which is very helpful in guiding our research. Following your suggestion, we analyzed the vertical wavelength and added the analysis to the Supplementary information. According to the results, we modified our discussions. We also thank you for the suggestion of estimating the energy transfer time scale. After studying it carefully, we think that using our observations from a single latitude through spectral analysis can hardly accurately estimate the amplitudes of the global transient waves. Therefore, we have not included this estimation in the current revision. Please find our detailed responses below in blue. We hope our revisions and responses have addressed all your concerns appropriately.

Reviewer Point 1.1. The authors use the resonant conditions of multi-wave nonlinear interaction to explain their observations. Namely, they suggest the process of SHG with conditions of frequency ($f_{i,i}=2f_i$), wavenumber ($k_{i,i}=2k_i$), and phase relations. General wave-wave interaction requires that all waves (initial and resulted) must satisfy the corresponding dispersion relations (for a two-wave interaction see e.g. Zaqqarashvili and Roberts, 2006). However, it is hard to be matched in the case of Rossby waves. The authors should show that the both waves satisfy the dispersion properties of Rossby waves. It is clear that 16-day initial wave satisfies the dispersion relation as second symmetric normal mode with $m=1$, but the similar discussion is absent for the SHG.

Reply: (Please also see our detailed explanation in response to Reviewer Point 1.2.)

The possibility of the SHG should not be a major concern of the current work because the possibility was demonstrated in the numerical simulation by *Pogoreltsev* (2001). Our analyses focused on the observational part, and we checked different observational perspectives and compared their consistency with theoretical expectations.

The reference in this comment reported an interaction between different types of waves. We point out this interesting fact in our revision.

Reviewer Point 1.2. The vertical structure of the wave modes is of vital importance to estimate the equivalent depth and consequently dispersion relations of the waves. The authors mentioned that the power spectrum is altitude-averaged. Is it possible to make, at least rough, estimation of the wave vertical structure and hence the equivalent depth? This may help the authors to answer the question 1.

Reply: in the revision, we carried out an analysis estimating the vertical wavelength, presented the it as the Supplementary, and discussed the results in Section 2. (Due to the limited data coverage in altitude, our estimation is associated with considerable uncertainties.) According to the results, we also modified our interpretation. Our revision explains the 16- and 8-day wave signatures in terms of atmospheric manifestations of Rossby wave normal modes.

The Reviewer suggests that both 16d and 8d waves should satisfy the dispersion properties of Rossby waves and seems to imply that each possesses a single equivalent depth and single vertical wavelength (L_z), as well as a single horizontal structure. However, this cannot be the case in the presence of realistic mean winds (especially for these low-phase-speed waves), which causes the actual atmospheric manifestations (AM) at these periodicities to project onto multiple modes. This causes the vertical phase structures to vary with latitude, meaning that the vertical wavelengths determined at our single latitude of 55N may not be globally representative. In mathematical terms, the system of equations is no longer

separable and no longer reducible to an eigenfunction-eigenvalue problem (i.e., a solution to Laplace's Tide Equation, LTE). Nevertheless, variables such as pressure, temperature, and geopotential should be able to be captured by a series of eigenfunctions (Hough functions), so there is some value in speaking in these terms. The following is an attempt to respond to the Reviewer in terms of the dispersion relations for Rossby waves, as requested.

The 16-day, $m = 1$ wave seen in atmospheric data is significantly distorted by the global mean wind field; in fact, during northern hemispheric winter, it is virtually excluded from the southern hemispheric middle atmosphere due to the easterly jet (e.g., *Forbes et al.*, 1995). It also occurs over a range of periods, roughly 12—20 days (e.g., *Forbes et al.*, 1995; *Forbes*, 1995). Note that the Rossby normal ("free") mode (NM) occurs at 12.5d. As shown in the attached dispersion diagram in Figure R1, for $m = 1$ (W1), the observed response at 16 days projects onto 4 Rossby modes with L_z of 32, 56, 86 km, and a mode with a very deep vertical scale. Each of these components has its own horizontal shape (Hough function) and vertical structure (determined by the equivalent depths). Taken together, they comprise the total observed 16-day wave, which has a horizontal structure that varies with height and a vertical structure that varies with latitude. The four positive equivalent depth modes noted above are the 1st symmetric and antisymmetric modes and the 2nd symmetric and antisymmetric modes, respectively. Because of the locations of these modes on the dispersion diagram, the observed 16-day wave as a whole is considered a Rossby wave.

As shown in the attached dispersion diagram in Figure R1, for $m = 2$ (W2), a similar situation exists for the 8d $m = 2$ waves. Note that the 8d $m = 2$ wave has the same zonal phase speed as 16d $m = 1$, and is likely to be similarly distorted by the zonal-mean wind field. Its AM projects onto Rossby waves with vertical wavelengths of 32 km and 72 km, and a near-NM with a deep vertical scale. These are the 1st symmetric, 1st antisymmetric and 2nd symmetric modes, respectively. Taken as a whole, due to the locations of these modes on the dispersion diagram and similarities with those associated with the 16d wave, the 8d $m = 2$ wave can also be termed a Rossby wave.

[redacted]

Reviewer Point 1.3. The wave amplitudes are important parameters in nonlinear interactions. The estimation of ratio between wave amplitude and phase speed of 16-day and 8-day waves may allow the authors to estimate the energy transfer time scale from 16 to 8-day oscillations. Then one can compare it to the observed time difference between the modes. This will enhance the confidence of reader in the interpretation suggested by the authors.

Reply: After studying your suggestion carefully, we think our observations from a single latitude through spectral analysis can hardly accurately measure the amplitudes of the global transient waves.

Therefore we have not included this estimation in the current revision.

As explained in response to Reviewer Point 1.2, our spectral analysis reveals the manifestations of planetary-scale normal modes at a given latitude, which can hardly represent the amplitudes of the global waves. Another consideration is associated with the transient properties of RWs. Experimental estimation of wave amplitudes through spectral analysis is subject to a predefined sampling window, and an accurate estimation entails stable waves persisting across the whole sampling window. However, RWs are transients and might not persist across our sampling windows. In addition, we can not reduce the sampling window size to improve the amplitude estimation because reducing the window in time will also reduce the spectral resolution in frequency. (According to Gabor's principle, the uncertainty tradeoff between temporal and spectral resolution.) With reduced frequency resolution, the amplitude and phase estimations of the 8-day wave are subject to contamination from, e.g., the 6-day wave.

Reviewer Point 1.4. Wave-wave interaction in the solar dynamo layer (references 5-6) is not yet confirmed to be a true process (though it is interesting). Therefore, I would advice to remove the discussion from the abstract but keep it in Introduction. But, of course, it is up to the authors.

Reply: Done. They are removed from the revised abstract.

Reviewer #2 (Remarks to the Author):

This paper reports a first observation of the excitation of a second harmonic of a Rossby wave in the middle atmosphere, in particular a wave with period ~ 16 days nonlinearly exciting one by resonance with half that period.

This observation is certainly interesting and worth reporting. The analysis methods are given in some detail, and the literature references are rather comprehensive. The authors well recognize the many places in the universe Rossby waves should be present.

That said, there are several ways the paper should be revised and improved before it is ready for publication.

Thank you very much for your constructive comments and suggestions. They are very valuable for improving our work. After studying your feedback, we have revised the manuscript accordingly. The main revisions are summarized here.

- (1) To investigate the novelty of our work, we made a broad literature study, tried to contact some meteorologists who work on tropospheric RWs, and got responses and discussions from two meteorologists. These efforts confirm the claimed novelty. Even so, we excluded "the first" claim in the revision.
- (2) We tried to explain our terminologies in plain language. A subsection was added to the Method to support the explanation of 'bicoherence'.

Below is a point-by-point response to your review report. All modifications in the manuscript have been highlighted in a pdf file enclosed. We hope our modifications and responses have appropriately addressed all your concerns.

Reviewer Point 2.1. The analysis terminology is very 'signal processing' oriented, as in EE, and, as such, will be difficult for geo and astrophysical readers to follow. More common language with brief intuitive descriptions understandable by a wider audience need to be included. Examples: 'cross-wavelet', 'bicoherence'

Reply: In the revision, we are trying to use plain language to explain the cross wavelet, by comparing it with the wavelet concept, and to interpret the bicoherence intuitively as a complex correlation defined between the energy-supplying and -receiving parties. Since this bicoherence interpretation is proposed for the first time, we revised the Method section and added two subsections to detail the interpretation.

Reviewer Point 2.2 More importantly, a first observation is claimed here, but there is little evidence in the text that the authors have checked the relevant literature for previous examples. It is not clear that these authors know the lower atmosphere literature well enough to be confident in this claim. It may be on more solid ground by comparison with other middle atmosphere analyses. Authors need to document how hard they looked, across what range of literature. It seems likely this phenomenon has been seen in the troposphere; Rossby waves there have been actively studied for at least 80 years.

Reply: Thank you for your comment.

First of all, in the revision, we modified the title, Abstract and Introduction by excluding the claim of "the first", which is also advised or requested by a GENERAL FORMATTING of the Journal's Guide at 3 Formatting instructions (nature.com). We also added a few relevant pieces of literature on theoretical analyses of the SHG of RWs in the ocean at the end of the Introduction.

Since the beginning of this study, we have made intensive efforts to search relevant papers. Still, we have not found any observational event of second harmonic generation (SHG) of Rossby waves (RWs). In response to this comment, we have contacted some meteorologists who work on tropospheric RWs. Two meteorologists, Professor Dehai Luo (Institute of atmospheric physics Chinese academy of sciences, Beijing, China) and Professor Antonio Speranza (Institute of Atmospheric Sciences and Climate, Bologna, Italy), replied to us and confirmed the novelty.

In the literature study on the tropospheric RWs, we found a terminology that sounds similar to SHG: the self-interaction of synoptic-scale RWs. This terminology was used in the hypothesis to explain the formation and maintenance of blocking events that have important implications for extreme cold in winter and heat waves in summer. After communicating with Professor Dehai Luo and Professor Antonio Speranza, we realized that this concept is not SHG. In SHG, a monochromatic wave generates a higher frequency wave, whereas the synoptic-scale RW self-interaction occurs between two

monochromatic waves from one package (e.g., Luo et al., 2014). In the latter, the frequency and wavenumber of one parent wave are close to those of the other, and the interaction gives rise to a secondary wave at a relatively very low frequency and very small zonal wavenumber. More importantly, we did not find any observational study trying to determine the evolution of the frequencies and zonal wavenumbers of waves participating in the synoptic-scale RW self-interaction. (To our understanding, this is because synoptic-scale RW activities typically involve multiple comparable monochromatic RWs at a time. Another reason is that the persistence and scale of the blocking are determined only by the frequency and wavenumber differences between the parent waves).

Compared with the synoptic-scale RWs, free planetary scale RWs in the troposphere play a relatively less important role in shaping the weather and, therefore, have attracted less attention. As demonstrated in response to Reviewer Point 2.3, the SHG phenomenon is rarely detectable in the middle atmosphere. (Only one is detected after scanning seven-year observations.) Since free planetary scale RWs' amplitudes are smaller in the troposphere than those in the middle atmosphere (e.g., Hirooka & Sciences, 2000), it is not surprising that this phenomenon has not been discovered in the troposphere.

Reviewer Point 2.3 Related to item 2, for readers to judge how important this observation is, they need to be given a better idea of how frequently such SHGs are likely to be present and detectable. Have the authors scanned a much longer sequence of synoptic maps or other data to estimate the frequency of occurrence? They need to present evidence of that. Detailed analysis of other cases is not necessary for this first study, but please give the reader some idea of how many times this behavior is likely to be found in much longer records. The importance of this behavior for simulation and especially prediction is tied to its frequency of occurrence.

Reply: In the revision, clarifications were added.

We detected only a single RW SHG event from seven-year (2013—2019) observations. The seven-year spectrum is attached here in Figure R2. Although the 16-day wave manifestation occurred in almost every winter, the 8-day manifestation occurred only once.

Figure R2, same plot as Figure 2 but in a 7-year window.

Reviewer Point 2.4 A common problem with language used in analyzing naturally occurring waves has to do with terminology like 'periods', 'frequencies' and 'wavelength'. 'Period' can mean the time it takes for an Rossby wave to propagate by its own wavelength, or how long it takes to propagate all the way around the spherical object, or it could relate to a periodic variation in the amplitude of the wave. The latter is particularly relevant when nonlinear couplings are present. The authors have certain definitions of these words in mind, and they may be consistent in their descriptions, but all these features are present in the data, so they need to distinguish carefully among them. 'Resonance' usually refers to an unusually large induced amplitude as well as period. Is the resonance reported here itself periodic in amplitude?

Reply: In this paper, "period" and "frequency" are used in three different contexts:

- 1) "wave period" refers to the time a wave takes for two successive crests to pass a specified point;
- 2) "spectral period/frequency" refers to the domain to which a time domain signal is converted through spectral analysis;
- 3) "rotation period" refers to the time that the object takes to complete a single revolution around its axis of rotation;

In the revision, these usages are discriminated and specified. The periodic variation in the wave amplitude is introduced in the revised Method. It is pointed out that this concept is beyond the scope of the current work.

"Wavelength" refers to the distance between two consecutive wave crests or between two consecutive wave troughs, which is also introduced in the revision.

"Resonance condition" refers to the phase matching relations in the interaction (e.g., *Gustavsson, 1975*). In the revision, we replace "resonance conditions" with "phase-matching relations" to exclude this confusion.

Reviewer Point 2.5 On a more detailed point, the identification of various types of modes discussed in lines 74-88 is too glib to be totally convincing; this discussion is key to the subsequent analyses. The words used, namely 'suggest', 'can be', 'might be' suggest the authors themselves have their doubts. More clarity on the uncertainty is needed here.

Reply: In the revision, we exclude these words. We also referred the readers to the Method Section for the uncertainty and assumptions of our data analysis.

Reviewer Point 2.6. There are places where the language does not make sense, or is scientifically inaccurate:

line 41: 'weathering' probably should be 'weather in'

Reply: Revised

line 51-53: the statement about detectability of Rossby waves with tides and gravity waves present applies perhaps in the upper or middle atmosphere, but not in the troposphere, where tides are much harder to detect in velocities compared to Rossby waves, and gravity waves are much more 'patchy' in their occurrence.

Reply: In the revision, we removed this problematic sentence.

line 68: what is 'round-based approach'? Probably should be 'ground-based approach'?

Reply: Revised.

Reviewer Point 2.7. In the solar context, the recently discovered 'tachocline nonlinear oscillations' (TNOs) should be mentioned, which also involve nonlinear mode mode interactions between Rossby waves of specific longitude wave number with themselves to produce changes in differential rotation and toroidal field ($m=0$ modes) amplitudes (Dikpati, Cally, McIntosh & Heifetz 2017, NatSR).

Reply: Done. Thank you for suggesting this reference. We have pointed out this interesting fact in the revision.

This Reviewer recommends revisions of the paper with attention to the points raised above. Happy to review the revised version.

Reviewer #3 (Remarks to the Author):

Review of the paper

"Discovery of Rossby wave second harmonic generation" by He and Forbes.

The manuscript, of course, is relevant, and may be of interest to a wide range of specialists in the field of not only atmospheric physics, but also, for example, astrophysics and even hydrodynamics.

Nowadays, not as many studies (as we would like to), are dedicated to the study of the generation of secondary (tertiary) waves in the atmosphere due to nonlinear interactions. In particular, similar studies are being carried out by our working group, with consideration of the balance equation of perturbed potential enstrophy, taking into account the rejection of the quasi-geostrophic approximation.

The work certainly deserves to be published.

However, I would like to note a few points to which the authors could pay attention.

We are grateful for your precious feedback. We have incorporated almost all your suggestions, which have led to significant improvements. The main revisions are:

- (1) we checked and rearranged the structure of the manuscript according to the Journal's guidelines;
- (2) one Figure, the revised Fig. 1, was added to sketch the theoretical foundations of Rossby waves;
- (3) one subsection was added to the Method to introduce all four possible secondary waves.

Please see below, in blue, for a point-by-point response to your comments and suggestions. In the submission, a pdf file is attached in which all revisions are highlighted in colour. We hope our revisions have addressed all your concerns properly.

Reviewer Point 3.1 Firstly, I am not very familiar with the requirements of this Journal, but the structure of the work seems strange to me. First, the results and conclusions are presented, and then the methods and approaches. Usually, it's the other way around. This makes it somewhat difficult to understand.

Reply: Thanks for your comment. The Editor has noted the current comment and requested us to arrange the manuscript following the guidelines on

<https://www.nature.com/documents/ncomms-formatting-instructions.pdf>

We revised the manuscript accordingly.

Next, I will refer to specific lines:

Reviewer Point 3.2 L27: The topic of solar and geomagnetic activity is not disclosed in the article. Therefore, I see no reason to announce it in the abstract.

Reply: It is removed from the abstract in the revision.

Reviewer Point 3.3 L35: I would like to take a closer look at the theoretical foundations of Rossby waves (and their importance for atmospheric dynamics), so that it is clear to a wide range of readers. Maybe even add a picture or scheme.

Reply: In the revision, we add one sketch to explain the theoretical foundations of Rossby waves and briefly introduce the importance of Rossby waves to geospace in the Introduction.

Reviewer Point 3.4 L51: weak amplitudes – this is only about troposphere.

Reply: We have removed this problematic sentence.

Reviewer Point 3.5 L106: As it was shown by (e.g., Longuet-Higgins, 1968, Pogoreltsev, 2001 – references are in the manuscript), if a signal consisting of two cosine waves with zonal wavenumbers and frequencies (m_1, a_1) and (m_2, a_2) pass through some quadratic system, the output from this system will contain the secondary waves $(2m_1, 2f_1)$, $(2m_2, 2f_2)$, $(m_1 + m_2, f_1 + f_2)$ and $(m_1 - m_2, f_1 - f_2)$. Eq (2) shows only summing. Or I didn't understand something.

Reply: In the revision, brief explanations were added in Section 3 which were detailed in a subsection added in the revision.

Suppose we constrain the frequency as non-negative; equation 2 shows only the secondary wave of the summing. Alternatively, one can define the frequency as a real value; then, Equation 2 notationally summarizes both the summing and differencing $\|f_1\| + \|f_2\|$ and $\|f_1\| - \|f_2\|$. We use the notation of real frequency in the Method to introduce the four secondary waves $(2m_1, 2f_1)$, $(2m_2, 2f_2)$, $(m_1 + m_2,$

$f_1 + f_2$) and $(m_1 - m_2, f_1 - f_2)$. Since the second harmonic generation involves only a special case of interaction with summing frequency, we constrain the frequency as non-negative in Section 3 for notational convenience.

Reviewer Point 3.6 In that context, you can also discuss the possible generation of an 8-day wave by other waves, for example, 5-day and 13-day ($m=1$), which is visible in your Fig.1

Reply: To generate an LSB of 8-day ($m=2$) entails parent waves of 13-day ($m=1$) and 5-day ($m=3$). The last requested wave cannot be found in Figure 1. Therefore, this Figure shows that the SHG of the 16-day wave is the only candidate.

Reviewer Point 3.7 L160: capture to Fig.1 remove "(a)" or add "a,b" to the Figure. Also, it would be much more convenient if you put Figure 1 and Table 1 as close to each other as possible.

Reply: Done.

References

- Forbes, J. M. (1995). Tidal and planetary waves. *Geophysical Monograph Series*, 87, 67–87. doi: 10.1029/GM087p0067
- Forbes, J. M., Hagan, M. E., Miyahara, S., Vial, F., Manson, A. H., Meek, C. E., & Portnyagin, Y. I. (1995). Quasi 16-day oscillation in the mesosphere and lower thermosphere. *Journal of Geophysical Research*, 100(D5), 9149. doi: 10.1029/94JD02157
- Gustavsson, H.-G. (1975, May). A Method of Analyzing the Resonance Conditions for a Three-Wave Interaction in a Magnetized Plasma. *Physica Scripta*, 11 (5), 319–322. doi: 10.1088/0031-8949/11/5/015
- Hirooka, T., & Sciences, P. (2000). Normal Mode Rossby Waves as Revealed by UARS / ISAMS Observations. *Journal of the Atmospheric Sciences*, 57(9), 1277–1285. doi: 10.1175/1520-0469(2000)057<1277:NMRWAR>2.0.CO;2
- Luo, D., Cha, J., Zhong, L., & Dai, A. (2014). A nonlinear multiscale interaction model for atmospheric blocking: The eddy-blocking matching mechanism. *Quarterly Journal of the Royal Meteorological Society*, 140(683), 1785–1808. doi: 10.1002/qj.2337
- Longuet-Higgins, M. S. (1968). The Eigenfunctions of Laplace's Tidal Equations over a Sphere. *Philosophical Transactions of the Royal Society A: Mathematical, Physical and Engineering Sciences*, 262 (1132), 511 – 607. doi: 10.1098/rsta.1968.0003
- Pogoreltsev, A. I. (2001). Numerical simulation of secondary planetary waves arising from the nonlinear interaction of the normal atmospheric modes. *Physics and Chemistry of the Earth, Part C: Solar, Terrestrial and Planetary Science*, 26(6), 395–403. doi: 10.1016/S1464-1917(01)00020-4

REVIEWERS' COMMENTS

Reviewer #1 (Remarks to the Author):

The authors replied to all my comments and revised the manuscript accordingly. Therefore, I'm happy to recommend the paper for publication.

Reviewer #2 (Remarks to the Author):

The authors have taken care of all my comments on the previous manuscript, and have revised accordingly. There are also other revisions, which are ok with me. I recommend acceptance of this paper in Nature Comm.

Reviewer #3 (Remarks to the Author):

The authors responded to all my comments. I recommend the manuscript for publication.